# Recovery of Bioactive Compounds from Unripe Red Grapes (cv. Sangiovese) through a Green Extraction

**DOI:** 10.3390/foods9050566

**Published:** 2020-05-04

**Authors:** Giovanna Fia, Ginevra Bucalossi, Claudio Gori, Francesca Borghini, Bruno Zanoni

**Affiliations:** 1Dipartimento di Scienze e Tecnologie Agrarie, Alimentari Ambientali e Forestali—DAGRI, University of Florence, Via Donizetti, 6, 50144 Firenze, Italy; giovanna.fia@unifi.it (G.F.); ginevra.bucalossi@unifi.it (G.B.); 2Vino Vigna, Via Claudio Monteverdi, 9, 50053 Empoli, Italy; c.gori@vinovigna.com; 3ISVEA SRL—Istituto per lo Sviluppo Viticolo Enologico Agroindustriale, Via Basilicata 1/3, Poggibonsi, 53036 Siena, Italy; f.borghini@isvea.it

**Keywords:** unripe grape extract, antioxidant activity, polyphenols, vitamins

## Abstract

Unripe grapes are a potential source of bioactive compounds which can exert antioxidant and antimicrobial activity. However, very little information is available about the composition of unripe grapes extracts and their extraction techniques. This study aims to evaluate the recovery of bioactive compounds and the antioxidant activity of the extract from unripe Sangiovese grapes obtained at an industrial-scale and the composition of the extract during processing. The extraction yield was approximately 75%. During the extraction, the total phenol (TP), antioxidant activity (AA) total anthocyanin (TAnt), pantothenic acid and choline content significantly increased. High concentrations of TP (2522 mgCATeq/L), AA (8227 μmolTEAC/L) and total water-soluble vitamins (1397 μg/L) were reached at the end of process. The antioxidant activity of the extract was positively correlated with all the phenol compounds and the highest correlations were found with procyanidin B1 (*r* = 0.994; *p*-value < 0.004), procyanidin B2 (*r* = 0.989; *p*-value < 0.004), kaempferol 3-*O*-glucoside (*r* = 0.995; *p*-value < 0.004) and quercetin 3-*O*-hexoside (*r* = 0.995; *p*-value < 0.071). Our findings contribute to the knowledge of the bioactive composition of unripe grapes. An efficient industrial-scale “green” extraction method, ready to be transferred to the wine sector, was developed to obtain a safe extract with a high concentration of bioactive compounds.

## 1. Introduction

Both red and white winemaking processes result in substantial amounts of solid organic waste [1]. Stalks, grape skins, grape pomace, grape seeds and wine lees undergo conventional treatments in order to reduce their environmental impact; common examples are the production of (i) compost from stalks, skins and pomace, (ii) spirits, organic acids and pigments from skins, pomace and lees, and (iii) oils from seeds [1,2]. The industrial wine sector is also exploring innovative solutions and developing marketable products which convert the industrial waste materials into by-products such as food ingredients and other products with high added value [3]. In particular, a renewed interest is being shown towards the extraction of bioactive compounds from grape by-products; indeed, their functional properties are currently of great interest in research, and in the food, nutraceutical and cosmetics industries [2,4,5,6,7].

One feasible by-product of the wine sector is the unripe grapes obtained from the thinning operations carried out to increase the quality of the wine produced. Cluster thinning is a common crop load adjustment practice that involves removing about one to two thirds of the clusters in order to promote the maturation of those clusters that remain on the vine [8,9]. Traditionally, unripe grapes are used to produce a semi-finished product called “verjuice”, which is a highly acidic juice made by pressing the unripe grapes. The “verjuice” is directly used as an ingredient in food seasonings or it is processed by cooking and adding vinegar and spices to obtain a “sour grape sauce” [10]. Thanks to its acidity and antioxidant activity, the “verjuice” also has some preserving properties, which are related to the organic acid and phenolic compound contents. The literature data has shown that it has an inhibiting effect on the enzymatic browning of fruit [11,12], an acidifying effect in food preparations [13,14,15], as well as a protecting one from fish lipid oxidation [16] and microbial activity [17,18].

Nevertheless, few studies have been performed on unripe grape bioactive compound extraction techniques [19,20]. Some authors optimized the temperature and ethanol/water solvent ratio for the extraction of phenolic compounds from unripe grapes at the lab-scale [21]. More recently, Fia et al. [22] used an innovative extraction machine in order to produce by maceration a liquid extract from unripe grapes. This extract was used after dehydration to increase the anti-browning capacity of white wines with promising results [22]. The above machine was also used to produce a powder extract from Merlot unripe grapes; functional and sensory proprieties of the extract were studied and the extract was used to fortify three vegetable food prototypes [23].

The aim of this study was to evaluate the recovery of bioactive compounds content and the antioxidant activity of the liquid extracts from unripe red grapes (cv. Sangiovese) during the different steps of a modified version of the above extraction by maceration (i.e., without the use of solvents) process. Since the biological properties of vitamins are well known and some of them are powerful antioxidants [24], the vitamins content of the liquid extracted from the unripe grapes was also studied for the first time.

## 2. Materials and Methods

### 2.1. Grapes

The red grape (cv. Sangiovese) samples from a commercial vineyard (Tenuta dello Scompiglio, Capannori, Lucca, Italy) were hand-thinned during the 2017 crop season for the industrial-scale processing of the liquid extracts from the unripe grapes. The unripe grape samples (approximately 1100 kg) were harvested on 21 August 2017, at the veraison ripening stage, and transported to the winery (Tenuta dello Scompiglio, Capannori, Lucca, Italy) in small cases (15 kg capacity).

### 2.2. Processing of Liquid Extracts

The liquid extract processing was carried out in the above winery according to the plans and the operating procedures in our previous papers [22,25], with the two following modifications: (i) more dry ice was added to prevent oxidative damage; and (ii) the pomace was pressed after the extraction phase to increase the extraction yield (Figure 1).

The total amount of 1080 kg of unripe grapes was discharged onto a conveyor belt and some dry ice was immediately added to decrease the temperature. Then the unripe grapes were crushed and destemmed. The crushed unripe grapes were transferred to the extraction by the maceration system [26], which was a thermostated stainless-steel tank (12500 L capacity) with four internal whorls for mixing. Some dry ice was added immediately and the crushed unripe grapes were mixed every 6 h for 30 min, for a total period of 96 h, at 6 °C. Some dry ice was added to the crushed grapes once a day before each pumping-over operation; about 1 kg of dry ice was added to every 3 kg of crushed grapes. After racking, the pomace underwent five pressing cycles, at increasing pressures, from 0.1 to 1.8 bars, in a membrane pneumatic press (Merlin model, Willmes GmbH, Lorsch, Germany). The liquid extract that was obtained from the pressing was mixed with the liquid extract obtained from the racking phase; all of the liquid extract was kept still inside the extraction system for 48 h at 6 °C for decantation and then it was filtered in order to remove any large particles (i.e., 1 mm or more in diameter).

The liquid extract samples were collected in triplicate during the extraction by maceration (i.e., 0, 24, 48, 72 and 96 h) and at each processing step after extraction (i.e., pressing, decantation and filtration). All the liquid extract samples were stored at −20 °C for the analyses.

### 2.3. Chemicals

All the chemicals were purchased from Sigma-Aldrich (Milan, Italy), except for the quercetin-3-O-glucoside, quercetin-3-O-glucuronide and rutin, which were supplied by Analytik GmbH (Rülzheim, Germany). The methanol and ethanol were supplied by Carlo Erba (Milan, Italy). Ultrapure water was obtained using a Thermo Scientific Milli-Q Gradient water purification system (Waltham, MA, USA).

### 2.4. Grape Chemical Characteristics and Phenolic Maturity

Randomized grape samples (about 3 kg) were collected from the vineyard (about 1.0 ha) and stored at −20 °C for further analysis. The chemical characteristics and phenolic maturity of the grapes were measured according to the literature [27]. One hundred berries were weighed and then pressed to separate the juice from the pomace. The juice was centrifuged at 2800× *g* for 10 min, and the sugar concentration, titratable acidity and pH of the clear juice were measured. The pomace was washed with water, then dried at 25 °C for 24 h, and weighed. The juice mass was calculated as the difference between the mass of berries and the mass of dry pomace. The pomace/juice (*P*/*J*) ratio was calculated as the mass of dry pomace/juice mass × 1000.

A further 100 grape berries were homogenized using a high-speed Ultra-Turrax at 11,500 rpm for 30 s. Fifty grams of homogenized grapes was extracted with a) 50 mL of 0.1 N HCl solution at pH 1.0 and b) 50 mL of tartaric acid solution (5 g tartaric acid in 20 mL 1 N NaOH, made up to a 1 L volume with distilled water) at pH 3.2. The samples were shaken at room temperature for 4 h, then they were centrifuged (13,440 G) at 4 °C for 10 min. The phenolic maturity parameters were determined through a spectrophotometric analysis of the juice. An ultraviolet–visible (UV–Vis) spectrophotometer (Lambda 10, Perkin Elmer, Italy) was used. The phenolic richness (*A280*) was obtained by measuring the absorbance at the wavelength of 280 nm, at pH 3.2, and was expressed as *absorbance units (AU) × dilution factor*. The anthocyanins extractable at pH 1.0 (*ApH1.0*) and pH 3.2 (*ApH3.2*) were measured at 520 nm and expressed as mg of malvidin-3-*O*-glucoside equivalents MAL eq/L of extract. The cellular maturity index (*EA%*) was calculated as *EA% = (ApH1 − ApH3.2)/ApH1 × 100*. The seed maturity index (*MP%*) was calculated as *MP% = [A280 nm − (ApH3.2/1000) × 40]/A280 nm*. The contribution of skin tannins to phenolic richness (*dTskin*) was calculated as *ApH3.2 × 40/1000* and the contribution of seed tannins to phenolic richness (*dTseed*) was calculated as *A280 nm—dTskin*.

### 2.5. Chemical Characterization of the Liquid Extracts

Samples of the processed liquid extract were filtered through a cellulose acetate membrane (0.45 µm diameter, Ministart Syringe Filter, Sartorius, Varedo (MB, Italy)). The measurements of the total phenol content, antioxidant activity, total anthocyanin content, intensity and hue colour were made on all the samples, while the measurements of the phenolic composition, glutathione and water-soluble vitamin content were only made at 0, 48, 96 h of extraction and after filtration, at the end of the process.

### 2.6. Total Phenol Content

One mL of extract was passed through a C18 Sep-pak cartridge (Waters, Milan, Italy) for phenol purification. The total polyphenol compound content was measured according to the Folin–Ciocalteau method [28] using a Perkin Elmer Lambda 10 spectrophotometer (Waltham, MA, USA). A standard curve was determined with (+)−catechin solutions at concentrations ranging from 5 to 500 mg/L by measuring the absorbance at 700 nm. The total phenol content (TP) was expressed as mg of (+)−catechin equivalents (CAT eq)/L of liquid extract.

### 2.7. Total Anthocyanin Content

The total anthocyanin concentration was measured according to the method described by Rigo et al. [29]. The samples were appropriately diluted with a solution (ethanol/water/hydrochloric acid) (70:29:1 *v*/*v*/*v*) at 0.12 M and the absorbance was measured at a wavelength of 540 nm, using a UV–Vis spectrophotometer (Lambda 10, Perkin Elmer, Italy). Malvidin-3-*O*-glucoside was used as a calibration standard and the total anthocyanin (TAnt) content was expressed as mg of malvidin-3-*O*-glucoside equivalents (MAL eq)/L of liquid extract.

### 2.8. Colour Intensity and Hue

The colour intensity (*CI*) and hue were measured using a 1 mm path-length quartz cell with distilled water as a reference [30]. A UV–Vis spectrophotometer (Lambda 10, Perkin Elmer, Italy) was used. The *CI* was expressed as the sum of the absorbance (*A*) at 420, 520 and 620 nm: *CI = (A420 + A520 + A620) × 10*. The hue (*H*) was expressed as the ratio between absorbance at 420 and 520 nm: *H = A420/A520*.

### 2.9. Phenolic Composition and Glutathione Content

The phenolic composition and glutathione content were measured with liquid chromatography-high-resolution mass spectrometry (LC-HRMS) following Fia et al. [22], using a chromatograph Accela 1250 (Thermo Fisher Scientific, Waltham, MA, USA) and a Kinetex F5 column (2.1 × 100 mm 1.7 µm—Phenomenex (Torrance, CA, USA)). An LTQ OrbitrapExactive mass spectrometer (Thermo Fisher Scientific, Waltham, MA, USA) was used. The quantitative analysis was performed with the TraceFinder™ 4.1 software (Thermo Fisher Scientific, Waltham, MA, USA) with an external standard method, using a linear regression from 0.05 to 1 g/L of five standard solutions.

### 2.10. Water-Soluble vitamin Contents

The vitamin analysis was performed via liquid chromatography-high-resolution mass spectrometry (LC-HRMS), according to the method described by other authors [31]. The samples were filtered to 0.45 µm and then they were analyzed. The liquid chromatograph was an Accela 1250 (Thermo Fisher Scientific, Waltham, MA, USA), equipped with a quaternary pump and a thermostatic autosampler. An Acquity BEH C18 column (2.1 × 100 mm 1.7 µm—Waters Milan, Italy) was used. The autosampler tray temperature was set at 10 °C and the column at 30 °C. Gradient elution was performed with water/0.1% formic acid (solvent A) and methanol/0.05% formic acid/5 mM ammonium formate (solvent B) at a constant flow rate of 400 µL/min, with an injection volume of 10 µL. An increasing linear gradient of solvent B was used. Separation was carried out in 12 min under the following conditions: 0 min, 2% B; 7 min, 20% B; 9 min, 20% B; and from 10 to 12 min, 2% B (modified from Owen et al. [30]). An LTQ Orbitrap Exactive mass spectrometer (Thermo Fisher Scientific, Waltham, MA, USA) equipped with an ESI source in positive mode was used to acquire the mass spectra in a full MS-data-dependent MS^2^ experiment. The operation parameters were as follows: source voltage, 3.5 kV; sheath gas, 40 (arbitrary units); auxiliary gas, 20 (arbitrary units); sweep gas, 0 (arbitrary units); capillary temperature, 300 °C; S-lens RF level, 70; automatic gain control (AGC) target, 1 × 106 for the MS mode and 1 × 105 for the MS^2^ mode. The samples were first analyzed in the full MS mode with the resolution set at 70,000, whereas the successive analyses were performed in the dd-MS^2^ mode with the resolution set at 17,500. An isolation width of 2 amu was used and the precursors were fragmented at a normalized collision energy of 30. The maximum injection time was set at 200 ms with one microscan, for both the MS mode and the MSn mode. The mass range was from m/z 150 to 1000. Quantitative analysis was performed with the TraceFinder™ 4.1. software (Thermo Fisher Scientific, Waltham, MA, USA) with an external standard method, using a linear regression of five standard solutions of a mix of all vitamins B and J from 0.05 to 1 g/L; instead, the calibration curve of vitamin C ranged from 1 to 5 g/L.

### 2.11. Antioxidant Activity

The antioxidant activity (AA) was determined by a 2,2-diphenyl-1-picrylhydrazyl (DPPH) spectrophotometric assay [32]. A Perkin Elmer Lambda 10 UV–Vis spectrophotometer (Waltham, MA, USA) was used. The decrease in absorbance was measured at 515 nm (maximum of DPPH absorbance). Trolox standard solutions were prepared daily in absolute ethanol at concentrations ranging from 10 to 600 µmol L^−1^. Antioxidant activity was expressed as µmol of the Trolox equivalent antioxidant capacity (TEAC)/L of the liquid extract.

### 2.12. Data Processing

A single industrial-scale processing of the liquid extracts was carried out; the liquid extract samples were collected in triplicate and all the measurements were carried out in triplicate. An analysis of the variance (two-way ANOVA—Least Significant Differences 5% level) was performed using the Statgraphics plus 3.1 software (The Plains, VA, USA) in order to assess the effect of time and replicates on the composition of the processed liquid extracts. A regression analysis was performed with the 19.02 version XLSTAT statistical software package (Addinsoft, GMSL Srl, Nerviano (MI), Italy) and the Pearson’s correlation coefficients (*r*) were calculated in order to assess the correlation between each measured variable and the antioxidant activity of the extract.

## 3. Results and Discussion

The chemical characteristics and phenolic maturity of the unripe grape sample picked at the veraison ripening stage during the 2017 crop season and used for the industrial-scale liquid extraction process are shown in Table 1. This degree of ripening was chosen since it is a good practice to thin Sangiovese grapes at the veraison stage in order to increase the wine quality [8]. The *P/J* ratio was 112.2, which means that the potential maximum amount of liquid extract from 100 kg of the unripe grapes sample was 89.3 kg; indeed, the percentage of dry pomace mass out of the unripe grape mass was 10% (Table 1). The cellular maturity index (*EA%*) was quite high (50.4%), indicating that anthocyanins were not easily extractable from the grape cells [27,33]. The contribution of the skin tannins to the phenolic richness (*dTskin* = 20.5) was higher than the contribution of the seed tannins to the phenolic richness (*dTseed* = 2.6); the seed maturity index value (*MP%* = 11.1) was consistent with the above behavior, showing seed lignification.

### 3.1. Recovery of Bioactive Compounds during Extraction by Maceration

The extraction system by maceration processed 1032 kg of crushed unripe grapes after the mechanical separation of the stems (48 kg—4.4% of the unripe grape sample) as shown in Figure 1. The applied extraction operating conditions can characterize the system as extraction by cold maceration. The cold maceration technique is widely used in winemaking with the purpose of improving the extraction of the phenolic compounds [34]. Moreover, the use of dry ice can protect the phenolic compounds from oxidation [35]. 

Figure 2 and Table 2 show the extraction kinetics of the total polyphenol compound content, total anthocyanin content, antioxidant activity, colour intensity and hue of the liquid extracts. The evolution of the phenolic compounds and anthocyanins extracted from the grapes reflected a typical solid–liquid extraction phenomenon [36]. An almost instantaneous dissolution of “free” solutes at the grape surface (i.e., leaching) was followed by the diffusion of the solutes from the interior of the grapes, tending towards an asymptote and reaching maximum values of approximately 2700 mg CAT eq/L and approximately 140 mg MAL eq/L, respectively. The antioxidant activity of the liquid extract increased linearly, reaching the approximate value of 4700 μmol TEAC/L after 96 h. The colour intensity increased significantly during the extraction; the hue value was initially high due to the poor anthocyanin content, then it decreased to less than 1.0 after 24 h and remained stable. The hue values and trend were consistent with the protection of the liquid extracts from oxidation during the extraction.

A variation of the phenolic profile occurred during the extraction (Table 3). With the exception of quercetin and myricetin, the phenolic compound contents showed a significant increase during the extraction (Table 3). At the beginning, the main phenolic compounds detected were phenolic acids and flavonols. The flavan-3-ol and procyanidin content was initially low; (−)−epicatechin and procyanidin B1 were detectable after 48 h and epicatechin-*O*-gallate was detectable after 96 h. At the end of the extraction, phenolic acids and flavan-3-ols were the most concentrated phenols in the liquid extract; caftaric acid, (+)−catechin and quercetin 3-*O*-glucuronide were the most abundant phenolic compounds. The phenolic profile reflected what has previously been highlighted by other authors on the composition of Sangiovese grapes [22,37]. The relevant extraction kinetics were influenced by the localization of the phenolic compounds in the grapes. The phenolic acids, the most abundant compounds in grape pulp, were extracted easily, while the flavan-3-ols and the procyanidins needed more time to be extracted due to the limit consisting of the thick cell wall of the seeds and skins where they are concentrated [19].

The total phenol content was correlated (*r* = 0.62; *p*-value < 0.004) with the antioxidant activity of the extract. This result is in agreement with those obtained by other authors [38,39]. The highest positive correlations were found between the antioxidant activity and procyanidin B1 (*r* = 0.994; *p*-value < 0.004), procyanidin B2 (*r* = 0.989; *p*-value < 0.004), kaempferol 3-*O*-glucoside (*r* = 0.995; *p*-value < 0.004) and quercetin 3-*O*-hexoside (*r* = 0.995; *p*-value < 0.071). Other authors found a significant correlation among the antioxidant activity and proanthocyanins, cathechins and procyanidins [40,41]. However, the antioxidant activity is strongly influenced by the grape variety and extraction method that both reflect the composition of the juice [42].

Reduced glutathione (GSH) was not measurable in the liquid extracts, whereas the enzymatic oxidative form, glutathione disulphide (GSSG), was present. 2-S-glutathionil caftaric acid (i.e., grape reaction product (GRP)), which is formed when o-quinones are in the presence of GSH [43], increased significantly during the extraction.

The water-soluble vitamin content increased from approximately 910 to approximately 1430 μg/L after 96 h of extraction (Table 4). At the start, pantothenic acid, choline, niacin and pyridoxine were already present in the liquid extract, indicating a leaching effect. During the extraction, the pantothenic acid and choline contents increased significantly, whereas the niacin and pyridoxine contents were stable. Information on the water-soluble vitamin profile of unripe Sangiovese grapes is lacking. Other authors described the water-soluble vitamins composition of many different cultivar. For example, Hagen et al. [42] observed pantothenic acid, pyridoxine and niacin in Chardonnay and Riesling berries at different stages of maturity. Recently, Andrade et al. [24] reviewed the composition of the bioactive compounds of different grape varieties (*Vitis vinifera*); water-soluble vitamins (pantothenic acid, vitamin B6, ascorbic acid, riboflavin and thiamine) were reported in red and green grapes, while, to the best of knowledge, data on choline were not available in the literature.

### 3.2. Recovery of Bioactive Compounds during Post-Extraction Steps

The extraction system was able to produce 692 kg of liquid extract after the final filtration step (Figure 1). Since the potential maximum amount of liquid extract from 1032 kg of crushed unripe grapes can be estimated at 920 kg (i.e., the grape *P/J* ratio was 112.2 as reported above), the liquid extraction yield was approximately 75%.

The bioactive compound content and antioxidant activity were measured during the post-extraction steps (Table 5). After pressing, the total polyphenol compound content and antioxidant activity increased significantly from 2407 to 2793 mg CAT eq/L and from 4738 to 8103 μmol TEAC/L, respectively; their relevant values were stable after decantation. A slight decrease in the total polyphenol compound content occurred after filtration (i.e., 2522 mg CAT eq/L), whereas the antioxidant activity was stable. The total anthocyanin content was stable, whereas a decrease occurred in the colour intensity; the slight decrease in the hue values suggested a low degree of oxidative damage during the post-extraction steps.

All of the liquid extracts showed a much higher phenolic content and antioxidant activity than those of the “verjuice” obtained by other authors and described in the literature. For example, other authors who studied unripe grape products found a total phenolic content of approximately 100–700 mg/L [10] and 1000–1500 mg/L [11], and antioxidant activity values of approximately 100–900 μmol TEAC/L [10] and 2000–4000 TEAC/L [11].

Some changes also occurred in the phenolic profile of the liquid extracts after filtration and after 96 h of extraction. The caftaric acid, coutaric acid, fertaric acid, (+)−catechin, (−)−epicatechin and procyanidin B1 and B2 contents were higher than those observed after 96 h of extraction (Table 3). The quercetin 3-*O*-glucuronide and 2-S-glutathionyl caftaric acid contents decreased significantly in the final extract, whereas the other phenolic compound contents were stable. The water-soluble vitamin content was stable; only the niacin content increased significantly in the final extract (Table 4).

## 4. Conclusions

In this study, an industrial-scale extraction process was positively tested to recover bioactive compounds from unripe grapes in the liquid extract. The extracted liquid had a high extraction yield, high phenolic compound and water-soluble vitamin contents and high antioxidant activity values compared with those measured in the “verjuice”, the traditional semi-finished product obtained from unripe grapes.

The strengths of the process were that (i) the method of extraction by maceration did not use solvents or preservatives, (ii) the method required a low temperature and caused little oxidative damage and (iii) the extraction yields were improved by the introduction of some post-extraction steps, which were innovations as compared with the previous version of the extraction by maceration process [22]. The process can be considered ready to be transferred to the wine sector in order to produce extracts, which can find applications as both functional ingredients to improve the health value of food and natural additives to protect food and beverages from oxidation.

## Figures and Tables

**Figure 1 foods-09-00566-f001:**
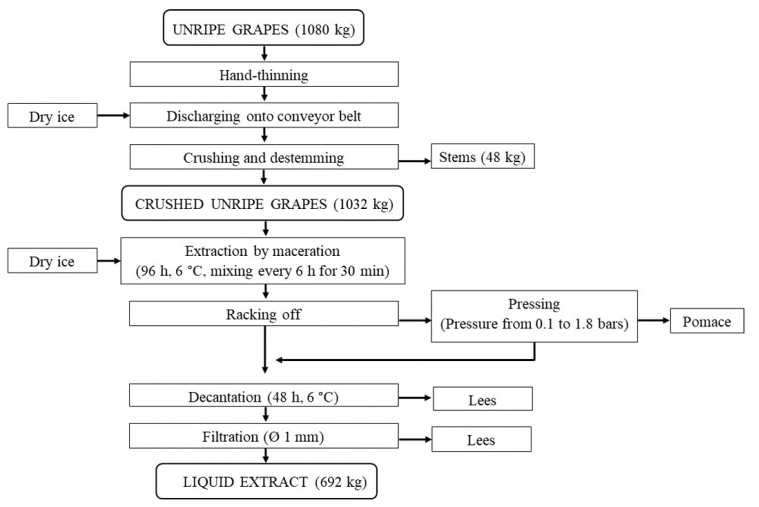
Flow-sheet of liquid extract processing from unripe grapes.

**Figure 2 foods-09-00566-f002:**
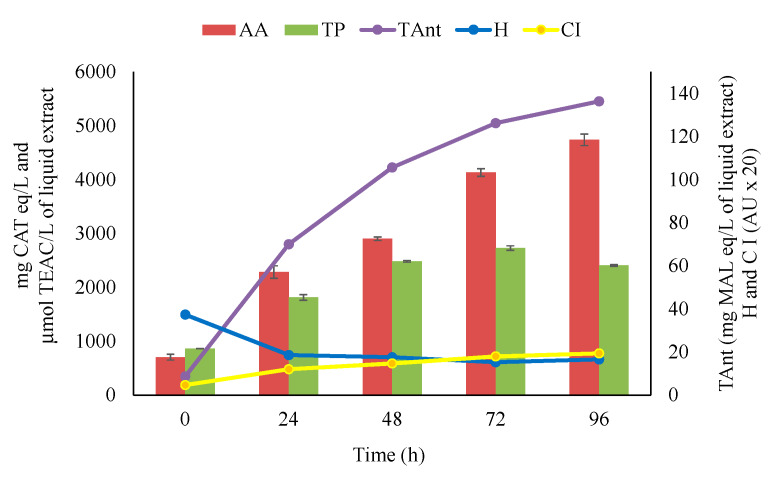
Antioxidant activity (AA), total phenol content (TP), total anthocyanins content (TAnt), hue (H) and colour intensity (CI) of the liquid extracts during the extraction phase. AU, absorbance Units. Values are the mean ± SD (*n* = 3). Bars represent standard error.

**Table 1 foods-09-00566-t001:** Chemical characteristics and phenolic maturity of the unripe grape sample.

*S*	*pH*	*TA*	*P/J Ratio **	*ApH1.0*	*ApH3.2*	*EA%*	*A280*	*Mp%*	*dTskin*	*dTseed*
143.0 ± 1.5	3.3 ± 0.02	5.7 ± 0.1	112.2 ± 4.0	1035.8 ± 30	513.5 ± 15	50.4 ± 1.6	23.1 ± 1.1	11.1 ± 0.8	20.5 ± 1.4	2.6 ± 0.03

*S* is the sugar content (g/L); *TA* is the titratable acidity (g tartaric acid/L); *P/J* ratio is the pomace/juice ratio (g dry pomace/g juice × 1000); *ApH1.0* is the anthocyanins extractable at pH 1.0 (MAL eq/L); *ApH3.2* is the anthocyanins extractable at pH 3.2 (MAL eq/L); *EA%* is the cellular maturity index; *A280* is the phenolic richness; *Mp%* is the seed maturity index; *dTskin* is the contribution of the skin tannins to the phenolic richness; *dTseed* is the contribution of the seed tannins to the phenolic richness. * The experimental data for the *P/J* ratio determination were as follows: berry mass = 99.1 g; dry pomace mass = 10 g; juice mass = 89.1 g.

**Table 2 foods-09-00566-t002:** Composition of liquid extracts during the extraction.

Variables	Time (h)				
	0	24	48	72	96
TP	864 ^d^	1813 ^c^	2483 ^b^	2728 ^a^	2407 ^b^
AA	702 ^d^	2282 ^c^	2902 ^b^	4134 ^a^	4738 ^a^
CI	0.2 ^e^	0.6 ^d^	0.7 ^c^	0.9 ^b^	1.0 ^a^
H	1.9 ^a^	0.9 ^b^	0.9 ^b^	0.8 ^c^	0.8 ^c^
TAnt	8.7 ^d^	70.0 ^c^	105.6 ^b^	126.3 ^a^	136.3 ^a^

AA, antioxidant activity (µmol TEAC/L); TP, total phenols (mg CAT eq/L); TAnt, total anthocyanins (mg MALqe/L); H, hue of colour and CI, colour intensity. Values are the mean ± SD (*n* = 3). Different letters represent significantly different values at *p*-value < 0.05.

**Table 3 foods-09-00566-t003:** Phenolic compounds and glutathione content of liquid extracts during extraction and after filtration at the end of the process.

Compound	Extraction			After Filtration
	0 h	48 h	96 h	
	Content (mg/L)			
*Phenolic acids*				
Caffeic acid	nd	nd	nd	nd
Caftaric acid	31.63 ± 1.98 ^d^	59.00 ± 2.89 ^c^	86.65 ± 4.49 ^a^	135.3 ± 3.36 ^a^
Coumaric acid	nd	nd	nd	nd
Coutaric acid	0.30 ± 0.05 ^d^	10.19 ± 0.18 ^c^	36.62 ± 0.41 ^b^	74.41 ± 3.49 ^b^
Ferulic acid	nd	nd	nd	nd
Fertaric acid	0.09 ± 0.01 ^d^	2.85 ± 0.07 ^c^	8.41 ± 0.06 ^b^	33.06 ± 0.96 ^a^
Gallic acid	nd	nd	nd	0.06 ± 0.00
*Flavonols*				
Quercetin	0.06 ± 0.06 ^a^	0.05 ± 0.00 ^a^	0.06 ± 0.00 ^a^	0.05 ± 0.00 ^a^
Quercetin 3-*O*-hexoside	0.72 ± 0.18 ^c^	4.89 ± 0.37 ^b^	12.86 ± 0.09 ^a^	12.33 ± 0.46 ^b^
Quercetin 3-*O*-glucuronide	1.59 ± 0.20 ^d^	8.99 ± 0.75 ^c^	32.51 ± 0.60 ^a^	23.54 ± 0.69 ^b^
Rutin	0.17 ± 0.04 ^c^	0.61 ± 0.05 ^b^	0.83 ± 0.03 ^a^	0.86 ± 0.03 ^a^
Isorhamnetin	nd	nd	nd	nd
Kaempferol	nd	nd	nd	nd
Kaempferol 3-*O*-glucoside	0.34 ± 0.01 ^c^	1.03 ± 0.04 ^b^	2.64 ± 0.05 ^a^	2.43 ± 0.14 ^a^
Myricetin	0.05 ± 0.00 ^a^	0.05 ± 0.00 ^a^	0.05 ± 0.00 ^a^	0.04 ± 0.02 ^a^
Myricetin-hexoside	nd	nd	nd	nd
Piceatannol	nd	nd	nd	nd
*Flavan-3-ols*				
(+)-Catechin	0.10 ± 0.03 ^d^	17.96 ± 0.12 ^c^	80.20 ± 1.18 ^b^	156.31±1.29 ^a^
(−)-Epicatechin	nd	0.49 ± 0.09 ^c^	7.35 ± 0.18 ^b^	17.21 ± 0.42 ^a^
Epicatechin-*O*-gallate	nd	nd	0.12 ± 0.01 ^a^	0.15 ± 0.01 ^a^
*Procyanidins*				
Procyanidin B1	nd	3.04 ± 0.05 ^c^	9.22 ± 0.08 ^b^	10.30 ± 0.15 ^a^
Procyanidin B2	0.19 ± 0.05 ^d^	6.79 ± 0.05 ^c^	9.22 ± 0.08 ^b^	17.71 ± 0.15 ^a^
*Glutathione*				
GSH	nd	nd	nd	nd
Grape reaction product-GRP	1.97 ± 0.16 ^c^	6.23 ± 0.81 ^b^	7.48 ± 0.92 ^a^	6.39 ± 0.75 ^b^
GSSG	1.08 ± 0.06 ^ab^	1.03 ± 0.05 ^b^	1.18 ± 0.08 ^a^	1.14 ± 0.09 ^ab^

GSH, reduced glutathione; grape reaction product-GRP, 2-S-Glutathionyl caftaric acid; GSSG, oxidized glutathione; nd, not detectable. Values are the mean ± SD (*n* = 3). Different letters indicate significantly different values among the columns (*p* ≤ 0.05).

**Table 4 foods-09-00566-t004:** Water-soluble vitamin content of liquid extracts during extraction and after filtration at the end of the process.

Water-Soluble Vitamins	Extraction			
	0 h	48 h	96 h	After Filtration
	Content (µg/L)			
Folic acid (B_9_)	nd	nd	nd	nd
Pantothenic acid (B_5_)	360.3 ± 10.4 ^b^	455.7 ± 24.8 ^ab^	492.0 ± 77.7 ^a^	452.0 ± 82.5 ^ab^
Ascorbic acid (C)	nd	nd	nd	nd
Biotin (B_7_)	nd	nd	nd	nd
Cobalamin (B_12_)	nd	nd	nd	nd
Choline (J)	400.3 ± 27.0 ^c^	603.3 ± 62.3 ^b^	779.7 ± 148.9 ^a^	782.3 ± 29.3 ^a^
Niacin (B_3_)	53.7 ± 5.9 ^b^	58.7 ± 5.5 ^ab^	51.0 ± 7.0 ^b^	62.0 ± 2.6 ^a^
Pyridoxal (B_6_)	nd	nd	nd	nd
Pyridoxamine (B_6_)	nd	nd	nd	nd
Pyridoxine (B_6_)	93.0 ± 12.1 ^a^	94.0 ± 7.0 ^a^	110.0 ± 7.9 ^a^	100.7 ± 8.4 ^a^
Riboflavin (B_2_)	nd	nd	nd	nd
Thiamine (B_1_)	nd	nd	nd	nd
Total content	906	1212	1433	1397

nd, not detectable. Values are the mean ± SD (*n* = 3). Different letters indicate significantly different values among the columns (*p* ≤ 0.05).

**Table 5 foods-09-00566-t005:** Composition of liquid extracts during post-extraction steps.

Parameter	End of Extraction	Post-Pressing	Post-Decantation	Post-Filtration
TP	2407 ± 30 ^b^	2793 ± 70 ^a^	2827 ± 30 ^a^	2522 ± 40 ^b^
AA	4738 ± 107 ^b^	8103 ± 793 ^a^	8130 ± 225 ^a^	8277 ± 731 ^a^
CI	0.97 ± 0.0 ^a^	0.91 ± 0.0 ^b^	0.89 ± 0.0 ^c^	0.77 ± 0.0 ^d^
H	0.83 ± 0.0 ^a^	0.75 ± 0.0 ^b^	0.83 ± 0.0 ^a^	0.73 ± 0.0 ^c^
TAnt	136.3 ± 10 ^a^	142.3 ± 16 ^a^	129.4 ± 4.0 ^a^	132.9 ± 4.0 ^a^

AA, antioxidant activity (µmol TEAC/L); TP, total phenols (mg CAT eq/L); TAnt, total anthocyanins (mg MALqe/L); H, hue of colour and CI, colour intensity. Values are the mean ± SD (*n* = 3). Different letters indicate significantly different values among the columns (*p* ≤ 0.05).

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
