# Peer review of "Recovery of Bioactive Compounds from Unripe Red Grapes (cv. Sangiovese) through a Green Extraction"

_foods, 2020, doi:10.3390/foods9050566_

Round 1

Reviewer 1 Report

The manuscript foods-776635 deals with the possibility of utilization of unripe grapes to produce high-value extract.

The authors used industrial-scale liquid extraction process (maceration) for the extraction of bioactive compounds from unripe grapes. Further more authors showed the kinetic of extraction of different phenolic compound during the extraction over different time periods. However, one qustion rises here; how the authors explain the increase in catechin, epicatechin and procyanidin  contents after filtration?

Overall, the paper is very well written and has a very good scientific approach. As I do not consider myself to be a native English speaker I cannot grade English. Finally, the manuscript is written according to the instructions for the authors and I can only suggest that the paper should adrress the question and provide quality graphics.

Other comments:

Figure 1 should be provide in better quality

 Author Response

Dear Reviewer,
Thank you for your positive judgment on our research paper. As you will see, our text has been corrected according to your comments as follows.
The measurement of phenolic compounds was done on the liquid extracts both at 0, 48, 96 h of extraction and at each processing step (see page 4, lines 127-130 of the manuscript, please). The increase of chatechin, epicatechin and procyanidin content detected after filtration (i.e. the last operation of our process) is actually due to the pressing operation, which preceded the filtration as you can check in the flow-sheet of the liquid extract processing (Fig. 1).
As you suggested, we improved the quality of Figure 1 by transforming it in a JPEJ extension before copying and pasting it in the Foods format.
Thank you for your cooperation.
Best regards,
Bruno Zanoni

Reviewer 2 Report

Research is well designed and the most significant part of it is that is set up at industrial-scale thus providing the producers valuable results which can be quickly transferred into production. Also, maceration process of unripe grapes is optimized and detailed chemical analysis was obtained also providing producers valuable results about quality of their product. The topic is scientifically relevant and interesting both to the scientific audience and also in the practice. The topic is averagely original since maceration is known and often used process (and in this research none of some new/modern processes were not examined), but valuable chemical analysis were made providing for the first time data about nutritional quality of "verjuice" product. Added value of this research is referenced in detail in answer 1. The paper is well written, understandable and all results are discussed. The text is clear and easy to read. Conclusions are based on the obtained results, evidences and arguments presented and proven by results. Conclusions address the main question, are derived and respond to set aim of the research.

I suggest just minor corrections in the tables name - shorten the main title while placing the table description below the table.

Author Response

Dear Reviewer,
Thank you for your positive judgment on our research paper. As you will see, our text has been corrected according to your comments as follows.
As you suggested, we made minor corrections in the Tables 1-5; the legends of the Tables were shortened, placing a part of description below them.
Thank you for your cooperation.
Best regards,
Bruno Zanoni

Reviewer 3 Report

Recovery of bioactive compounds from unripe red grapes (Sangiovese cv.) through a “green” extraction

I write you in regards to manuscript # foods-776635 entitled " Recovery of bioactive compounds from unripe red grapes (Sangiovese cv.) through a “green” extraction" which you submitted to the foods.

Authors need to follow the following instructions to improve this manuscript

  • Title: Recovery of bioactive compounds from unripe red grapes through a green extraction. The author can skip others from the title.
  • The abstract should rewrite based on the best result of this experiment. Include data of each parameter and correlation value (r =? and significant level =?)
  • Should discard unwanted words, especially, background, methods, results, conclusions
  • It is the first time; I saw that cultivar wrote like Sangiovese Please write a cv. Sangiovese or ‘Sangiovese.' Do not make Italic. Correct it from the whole manuscript.
  • Page 2, Line 73: the veraison ripening stage (i.e., 21 August 2017). Author mention here ripening stage but indicated date and year in the bracket
  • Page 2, Line 76-79: The author already published two papers (22,25). Just two modified. There is no novelty in this experiment.
  • Page 3, Figure 1: Mentioned industrial scale. But ---
  • Page 4, Line 106: The author used only one hundred grams. There is no relevance. The author should follow one procedure. They can use a small quantity from a large sample.
  • Page 4, Line 125-130: Method should check carefully and use a reference with the instrument
  • Page 4, Line 132-137: spectrophotometer. Is it correct? Confirm.
  • Page 4, Line 146-149: Is it correct? Confirm. Mention the instrument name, company, and country
  • Page 5, Line 160-183: very complex. There are many simple methods.
  • Page 5, Line 185-188: mention ----nm. Mention the instrument name, company, and country
  • Page 6, Line 199: Sangiovese Words no need to use again and again
  • Result and Discussion should rewrite based on the best findings with relevant references.
  • Conclusion: The author should rewrite based on the best result of this experiment.
  • English correction is mandatory from a native speaker or English proofreading company
  • Please check carefully before resubmission.

I recommend to improve the manuscript and resubmit.

Author Response

Dear Reviewer,

Thank you for giving me the opportunity to improve our work and make it clearer through your comments. As you will see as follows, our text has been corrected, and hopefully exhaustive responses have been given.

Thank you for your cooperation.

Best regards,

Bruno Zanoni

- Title: The title as changed in “Recovery of bioactive compounds from unripe red grapes (cv. Sangiovese) through a green extraction”. If the reviewer is agree, the authors would rather leave the the word “cv. Sangiovese” in the title.

- Abstract: the authors deleted unwanted words (i.e. background, methods, results, conclusions) from the abstract and they partially rewrote it, including data of each parameter and correlation value (r and p-values) as follows: “Unripe grapes are a potential source of bioactive compounds which can exert antioxidant and antimicrobial activity. However, very little information is available about the composition of the unripe grapes extracts and extraction techniques. This study aims to evaluate the recovery of bioactive compounds and the antioxidant activity of the extract from unripe Sangiovese grapes obtained at an industrial-scale and the composition of the extract during processing. The extraction yield was approx. 75%. During the extraction, the total phenol (TP), antioxidant activity (AA) total anthocyanin (TAnt), pantothenic acid and choline content significantly increased. A high concentration of TP (2522 mgCATeq/L), AA (8227 molTEAC/L) and total water-soluble vitamins (1397 g/L) was reached at the end of process. The antioxidant activity of the extract was positively correlated with all the phenol compounds and the highest correlations were found with procyanidin B1 (r = 0.994; p-value <0.004), procyanidin B2 (r = 0.989; p-value <0.004), kaempferol 3-O-glucoside (r = 0.995; p-value <0.004) and quercetin 3-O-hexoside (r = 0.995; p-value <0.071). Our findings contribute to the knowledge of the bioactive composition of unripe grapes. An efficient industrial-scale “green” extraction method, ready to be transferred to the wine sector, was developed to obtain a safe extract with a high concentration of bioactive compounds.”

The p-values were also added in the text at page 7, lines 257-259.

- The Authors wrote “cv. Sangiovese” and corrected it from the whole manuscript.

- Page 2, Line 73. For clarity, the authors rephrased the sentence as follows: “The unripe grape samples (approx. 1100 kg) were harvested on 21 August 2017, at the veraison ripening stage,….”.

- Page 2, Lines 76-79. The authors published only one paper (22) on this topic; the other cited reference (25) is an European Patent about a multi-functional oenological machine. This paper submitted to Foods journal is an up-grade of the previous one; the novelty regards the addition of two steps (i.e. discharging of grapes on a conveyor belt with dry ice addition and pressing of pomace) in the processing flow-sheet. Moreover, unlike in the previous work (22), analyses were made step-by-step through all over the production process. The novelty is also related to the chemical characterization of the unripe grapes and unripe extract composition, especially regard to the water-soluble vitamin profile. In fact, few data are available on this topic. The new procedure is also set up at an industrial-scale and it is ready to be transferred into production.

The authors have tried to clarify the novelty of experiment improving the aim (“…of a modified version of the above extraction by maceration (i.e., without the use of solvents) process”) and the conclusions (“…which were innovations as compared to the previous version of the extraction by maceration process [22]”) of the paper.

- Page 3, Figure 1. The authors were not able to understand completely the reviewer’s comment “Figure 1: Mentioned industrial scale. But ---“. The authors believed that an approx. 1000 kg unripe grapes samples can be assumed as a good example for a processing in an actual winery organization. However, the title of the 2.2 paragraph of the Materials and Methods section was changed.

- Page 4, line 106. The authors added the missing information about the large sample of grapes picked for the analysis by adding the following sentence: “Randomized grape samples (about 3 kg) were collected from the vineyard (about 1.0 ha) and stored at -20 °C for further analysis.” The authors added in the text information about the instrument used: “An ultraviolet-visible (UV-Vis) spectrophotometer (Lambda 10, Perkin Elmer, Italy) was used.”

- Page 4, lines 125-130. In these lines the authors specified which analyses were made on the different samples taken during the production of the extract. In addition, the authors described here how the samples were treated by filtration prior to analysis. No instrument is mentioned yet and therefore there is no need for a reference. Then, for clarity, the authors made the following modification to the text:

2.5. Chemical characterization of the liquid extracts

Samples of the processed liquid extract were filtered through a cellulose acetate membrane (0.45 μm diameter, Ministart Syringe Filter, Sartorius, Varedo (MB, Italy)). The measurements of the total phenol content, antioxidant activity, total anthocyanin content, intensity and hue colour were made on all the samples while the measurement of the phenolic composition, glutathione and water-soluble vitamin content were only made at 0, 48, 96 h of extraction and after filtration, at the end of process.”

- Page 4, lines 132-137. The authors confirm that the instrument used to measure the absorbance of phenol compounds according to Folin-Ciocalteau method was a spectrophotometer. The authors added in the text the information about the wavelength at which the measure was made: “A standard curve was determined with (+)-catechin solutions at concentrations ranging from 5 to 500 mg/L by measuring the absorbance at 700 nm.”

- Page 4, Lines 146-149. The authors confirm that it is correct. The authors added in the text information about the instrument as follows: “An UV-Vis spectrophotometer (Lambda 10, Perkin Elmer, Italy) was used.”

- Page 5, Lines 160-183. The authors are agree with the reviewer that some methods are simpler that their applied method. However, the authors found it impossible to address the above comment, since they believed that their particular method was the most appropriate for the experiment.

- Page 5, Lines 185-188. The authors added the information about both the wavelength, at which the measure was made, and the instrument name, company, and country: “A Perkin Elmer Lambda 10 UV-Vis spectrophotometer (Waltham, MA, USA) was used. The decrease in absorbance was measured at 515 nm that is the maximum of DPPH absorbance.”

- Page 6, Line 199. The authors deleted the unwanted words.

- Results/Discussion and Conclusions should rewrite…… The authors were not able to understand completely the reviewer’s comment, since no specific remarks were done by the reviewer. However, the authors tried to realize the reviewer’ comments, improving the above sections, and they included the following new information and comments about the antioxidant power and phenolic profile of the samples: “ The total phenol content was correlated (r = 0.62; p-value <0.004) with antioxidant activity of the extract. This result is in agreement with those obtained by other authors [40,41]. The highest positive correlations were found between the antioxidant activity and procyanidin B1 (r = 0.994; p-value <0.004), procyanidin B2 (r = 0.989; p-value <0.004), kaempferol 3-O-glucoside (r = 0.995; p-value <0.004) and quercetin 3-O-hexoside (r = 0.995; p-value <0.071). Other authors found significant correlation among antioxidant activity and proanthocyanins, cathechins and procyanidins [42,43]. However, the antioxidant activity is strongly influenced by the grape variety and extraction method that both reflect the composition of juice [43].

  1. Breksa, A.P. III; Takeoka, G.R.; Hidalgo, M:B.; Vilches, A.; Vasse, J.; ramming, D.W. Antioxidant activity and phenolic content of 16 raisin grape (Vitis vinifera L.) cultivars and selections. Food Chem. 2010, 121, 740-745.
  2. Orak, H.H. Total antioxidant activities, phenolics, anthocyanins, polyphenoloxidase activities of selected red grape cultivars and their correlations. Sci. Hortic. 2007, 111, 235-241.
  3. Bartolomé, B.; Nuñez, V.; Monagas, M.; Gómez-Cordovés, C. In vitro antioxidant activity of red grape skins. Eur. Food Res. Technol. 2004, 218, 173-177.
  4. Lingua, M.S.; Fabani, M.P.; Wunderlin, D.A.; Baroni, M.V. From grape to wine: Changes in phenolic composition and its influence on antioxidant activity. Food Chem. 2016, 208, 228-238.

- English correction is mandatory from a native speaker or English proofreading company. Before submitting to Foods journal, the Authors had the manuscript checked by a native English speaking professional. Anyway, as suggested by the Reviewer, the Authors checked again both language and style.

Thank you for your cooperation.

Best regards,

Bruno Zanoni

Round 2

Reviewer 3 Report

The manuscript is perfect.